# CLQ: CROSS-LAYER GUIDED ORTHOGONAL-BASED QUANTIZATION FOR DIFFUSION TRANSFORMERS

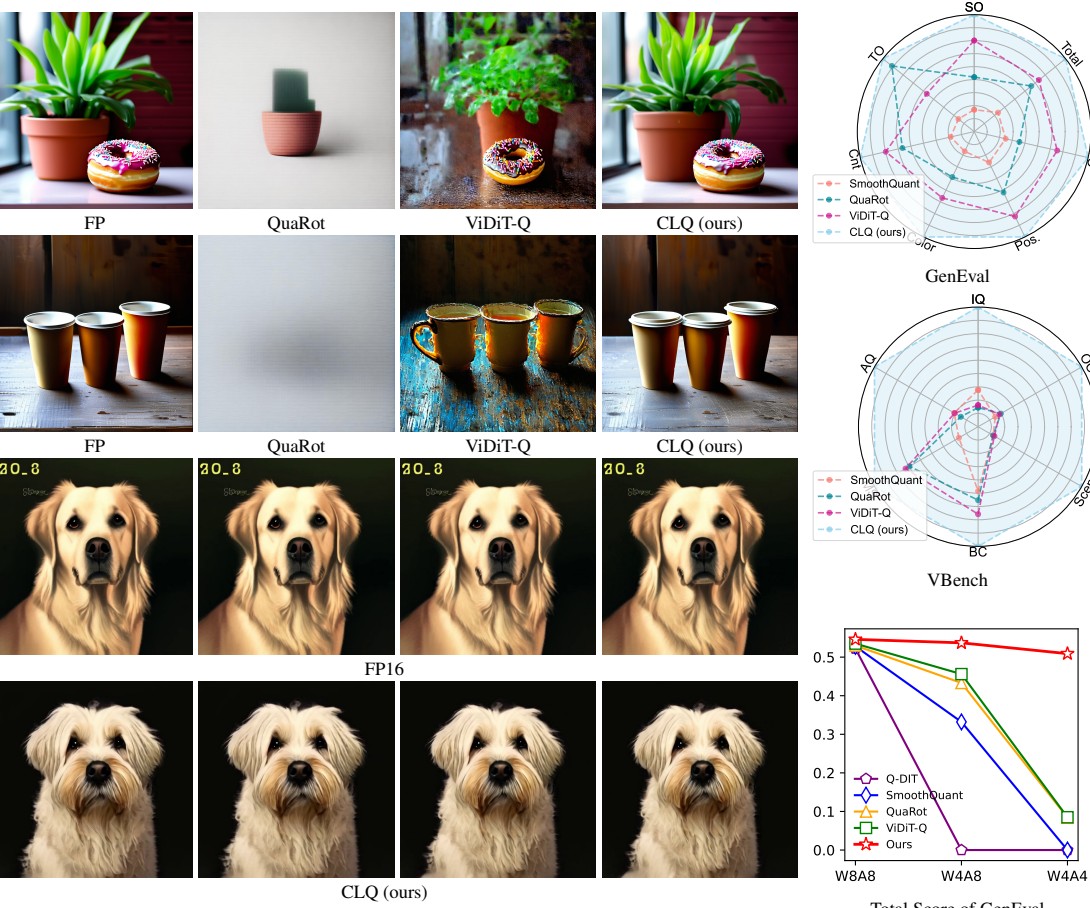

Figure 1: CLQ is a post-training quantization framework for DiTs on visual generation tasks. CLQ could compress the DiTs into W4A4 while preserving the high-quality output of the original model.

## ABSTRACT

Visual generation quality has been greatly promoted with the rapid advances in diffusion transformers (DiTs), which is attributed to the scaling of model size and complexity. However, these attributions also hinder the practical deployment of DiTs on edge devices, limiting their development and application. Serve as an efficient model compression technique, model post-training quantization (PTQ) can reduce the memory consumption and speed up the inference, with inevitable performance degradation. To alleviate the degradation, we propose **CLQ**, a cross-layer guided orthogonal-based quantization method for DiTs. To be specific, **CLQ** consists of three key designs. First, we observe that the calibration data used by most of the PTQ methods can not honestly represent the distribution of the activations. Therefore, we propose cross-block calibration (CBC) to obtain accurate calibration data, with which the quantization can be better guided. Second, we propose orthogonal-based smoothing (OBS), which quantifies the outlier score

of each channel and leverages block Hadamard matrix to smooth the outliers with negligible overhead. Third, we propose cross-layer parameter searching (CLPS) to search. We evaluate **CLQ** with both image generation and video generation models and successfully compress the model into W4A4 with negligible degradation in visual quality and metrics. **CLQ** achieves 3.98x memory saving and 3.95x speedup with real-world deployment testing. Our code will be released soon.

# 1 INTRODUCTION

Recent advances in visual generation have demonstrated remarkable progress in producing high-quality and photorealistic images and videos (Singer et al., 2022). In particular, diffusion-based models have rapidly become the dominant paradigm in terms of fidelity, diversity, and controllability. More recently, diffusion transformers (DiTs) (Peebles & Xie, 2023) have further pushed the frontier by combining the strong generative capacity of diffusion processes. With the scalability and representation power of transformer architectures, DiTs have enabled impressive achievements across various applications, including image synthesis (Chen et al., 2023b), video generation (OpenAI, 2024), and multimodal content creation (Deng et al., 2025). However, the superior performance of DiTs comes at the cost of massive model size and computational complexity. As these models continue to scale, merely inference becomes prohibitively expensive, limiting their deployment in real-world scenarios. To generate a $512 \times 512$ video with only 16 frames, the OpenSORA (OpenAI, 2024) model, as an example, takes more than 50 seconds on an NVIDIA A100 GPU and consumes over 10GB.

Model Quantization (Jacob et al., 2018), as an indispensable step in deployment, can effectively compress model storage and accelerate inference. Converting high-bit-width floating-point (FP) numbers into low-bit-width integers (INT), model quantization can compress models by several times. Moreover, considering hardware architecture, the integer operations are usually simpler and more efficient than floating-point operations (Zhang et al., 2021), thereby allowing quantization to save bandwidth and accelerating computation (Liu et al., 2025). However, conversion from high-bit-width to low-bit-width also inevitably brings performance degradation due to the quantization error. Specifically, when considering DiTs in visual generation, quantization research is still underexplored (Wu et al., 2025). Previous research mainly focuses on high-bit quantization and suffers from extreme performance degradation when compressing to a lower bit-width (Zhao et al., 2025).

Specifically, when compressing to a lower bit-width, these methods usually suffer from the outliers in the network (Xiao et al., 2023). When considering the distribution, both weight and activation obey the bell distribution. Previous research demonstrates that the outliers' impact on the performance is negligible (Liu et al., 2024a). However, it is not the case for DiTs. We observe that there are more extreme values in DiTs compared to low-level Transformer models. These extreme values exert a substantial influence during model quantization. First, they introduce significant rounding errors. A common approach to determine quantization boundaries is through the use of min–max or percentile methods (Liu et al., 2024b). Under both schemes, the resulting quantization ranges tend to be excessively large in absolute value, which amplifies rounding errors and thereby degrades model performance. Second, if one chooses to clip these extreme values, model performance can also be severely affected. This is because such extreme values often encode critical information that is essential to the generative process. Therefore, how to properly handle extreme values remains an open problem, as it has a profound impact on model performance.

In this work, we propose **CLQ** to compress visual generation models to ultra-low bit-width while preserving the high-quality output of the original model, as shown in Fig. 1. First, we begin with the analysis of the calibration process. Previous methods use the full-precision input and output of each module during calibration. However, quantization error accumulates as a discrepancy exists between the full-precision input data and the quantized ones. Therefore, we propose a novel cross-block calibration (CBC) method to obtain more accurate calibration data, thereby providing precise guidance in the following quantization process. Second, we propose orthogonal-based smoothing (OBS). OBS first detects the uneven channel and sorts the channels with an outlier score. Then, OBS smooths uneven activations and weight matrices using a block Hadamard transform, preventing outliers in irregular channels from affecting the more stable ones. In the quantization phase, we novelly propose cross-layer parameter searching (CLPS), which analyzes the most influenced layers and leverages cross-layer to obtain the quantization parameters with the minimum quantization error.

Combining both CBC, OBS, and CLPS, our proposed CLQ allows the model to still enjoy almost lossless performance. Moreover, when compressed into 4 bits, the model has a speedup of $3.95\times$, making it more applicable in real-world applications and deployment.

We summarize our contributions as follows:

- We propose cross-block calibration (CBC), a novel method for calibration data collection. CBC could provide accurate calibration data and minimize the accumulated quantization error, improving the quantized model's performance.

- We propose orthogonal-based smoothing (OBS), which leverages rotation matrices to smooth the outliers and Hadamard matrices to be calculation-efficient.

- We propose cross-layer parameter searching (CLPS), which searches the quantization parameters with the second-order norm of the cross-layer output.

- We conduct extensive experiments to evaluate the proposed **CLQ** in visual generation tasks. We achieve W4A4 compression in visual generation and a 3.95x speedup ratio with almost lossless model performance, pushing visual generation closer to real applications.

## 2 RELATED WORK

### 2.1 VISUAL GENERATION

Visual generation has progressed from early GAN-based (Goodfellow et al., 2020) methods to more stable diffusion models (Rombach et al., 2022). GANs, such as DCGAN (Radford et al., 2015) and StyleGAN (Karras et al., 2019), achieved impressive image realism but suffered from mode collapse and training instability. Variational autoencoders (Kingma & Welling, 2013) provided stable optimization but lower fidelity. Diffusion models changed the landscape, starting with DDPM (Ho et al., 2020) and later improvements like DDIM (Song et al., 2020) and classifier-free guidance (Ho & Salimans, 2022). These methods achieved state-of-the-art performance in image and video generation, showing robustness and controllability that surpassed previous paradigms.

### 2.2 DIFFUSION TRANSFORMER

The backbone of diffusion models was initially a U-Net architecture (Ronneberger et al., 2015). Recent works (Rombach et al., 2022) replaced U-Nets with Transformers (Vaswani et al., 2017) to improve scalability and representation. DiT showed that pure transformers could outperform convolutional backbones in diffusion tasks. U-ViT further validated the effectiveness of hierarchical transformer designs for generation. These architectures enabled stronger scaling laws, similar to large language models, and unlocked new applications in high-resolution image synthesis and multi-modal generation. Transformers thus became a promising backbone for diffusion-based visual generation.

### 2.3 POST-TRAINING QUANTIZATION

Quantization (Jacob et al., 2018) has become a practical approach for compressing deep networks without retraining. Early PTQ methods, such as percentile quantization (Li et al., 2019), achieved efficiency but limited accuracy. Advanced PTQ methods like GPTQ (Frantar et al., 2022) and DuQuant (Lin et al., 2024) improved precision for large language models by handling activation outliers and optimizing quantization error. SmoothQuant Xiao et al. (2023) leverages diagonal matrices to smooth activation and transfer the quantization difficulties from activations to weights. ViDiT-Q Zhao et al. (2025) presents a successful practice for PTQ on visual generation tasks. However, these current PTQ methods collapse when it comes to ultra-low bit-width, such as W4A4. One way to compensate for the quantization loss is to use finer quantization granularity, which also slows the inference. Therefore, applying PTQ to DiT models remains an open question. This motivates research on PTQ tailored for DiTs, especially for visual generation tasks.

## 3 METHOD

### 3.1 PRELIMINARY

Post-training quantization (PTQ) reduces model precision without retraining. A common practice is asymmetric uniform quantization, which achieves a trade-off between hardware ecosystem and

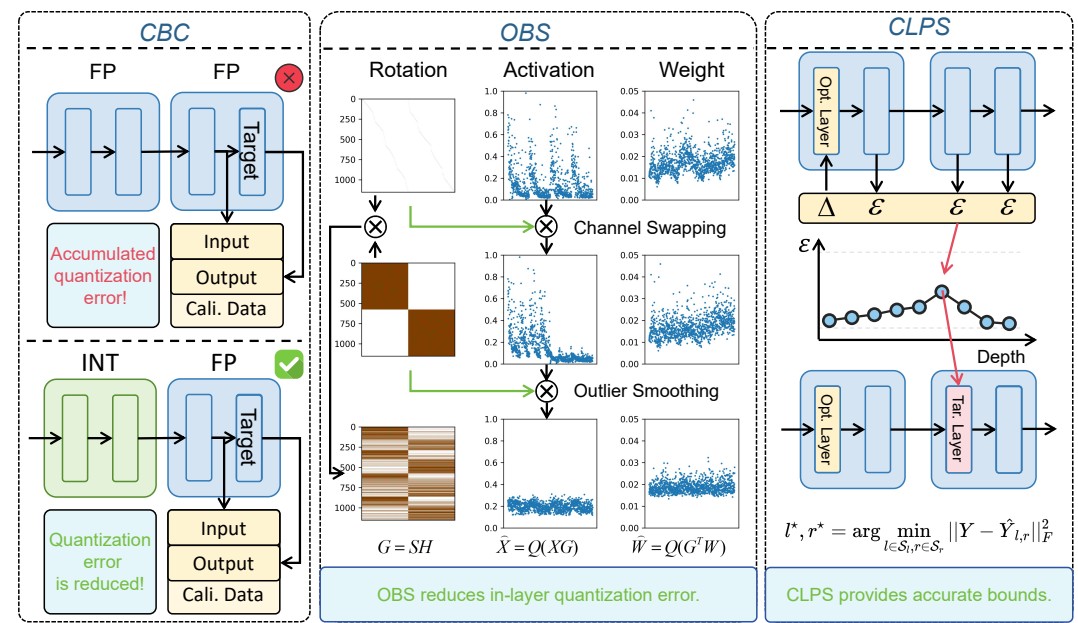

Figure 2: The overall pipeline of the proposed CLQ, which consists of three novel designs. Cross-block calibration provides accurate calibration data and reduces the accumulated quantization error. Orthogonal-based smoothing leverages column-swapping matrices and block Hadamard matrices to smooth the outliers with negligible overhead in inference and storage. Cross-layer parameter searching analyzes the sensitive target layer and reduces the cross-layer quantization error. All three methods together guarantee the outstanding performance of the proposed CLQ.

model performance. Given a full-precision weight $W \in \mathbb{R}$, it is mapped into a quantized value $\hat{W}$ by

$$\hat{W} = \mathrm{Q}(w, l, r, n) = \mathrm{Round}\left(\frac{\mathrm{clip}\left(W, l, r\right) - l}{r - l} \cdot (2^n - 1)\right) \cdot \frac{r - l}{2^n - 1}, \tag{1}$$

where $l$ and $r$ are left and right bounds for quantization, $n$ is the number of bits, clip$(\cdot)$ clamp the input value into the given range, and Round$(\cdot)$ rounds the value into the nearest integers. Both weights and activations can be quantized in this manner. Previous works set $l = \min(w), r = \max(w)$ directly, which is not appropriate for DiTs. In our work, $l$ and $r$ are the quantization parameters to be optimized. $n$ is assigned as a fixed given value. To be brief, W4A8 is short for quantizing the weight into 4 bits and activation into 8 bits. We adopt the dynamic quantization scheme, where the quantization parameters for activations are calculated dynamically according to the input, and the weights are only quantized once.

## 3.2 CROSS-BLOCK CALIBRATION

In the PTQ process, a critical step is collecting data to analyze the distribution of model activations, which subsequently guides the optimization of the model's quantization parameters. This procedure is referred to as calibration, and the data used is known as the calibration set. Previous approaches typically involve collecting full-precision activations across the entire network at once and calibrating the quantization parameters using full-precision inputs and outputs. Our analysis below reveals that this method leads to the layer-by-layer accumulation of quantization errors, which ultimately impacts the model's performance.

Specifically, previous methods leverage the floating-point input and output as the collected calibration data. However, quantization compresses the weight of all the layers, leading to a mismatch between the quantized weights and the original ones. Moreover, considering that the input of the target layer is the output of the previous layer, the input has changed when calibrating the target layer, as the previous layer's output changes after quantization. Further, the changes of the input accumulate as the calibration goes deeper, leading to a significant calibration error.

To address this problem, we propose cross-block calibration (CBC) to obtain accurate calibration data, with which the quantization can be better guided. First, when collecting the calibration data,

all previous layers must be properly quantized. Here, "properly" refers to adopting the proposed PTQ method. Only then can we reduce the quantization error to the best extent. Considering the limited depth of one transformer block, we choose a coarser granularity for easier implementation. To be specific, when collecting the target layer's calibration data, we only require that the previous transformer blocks is quantized instead of all the layers ahead.

## 3.3 Orthogonal-Based Smoothing

Smoothing the model through rotation matrices has been proven to be an efficient method (Liu et al., 2024b). However, previous studies (Lin et al., 2024; Ashkboos et al., 2024) typically adopted dynamic approaches to construct the rotation matrix, which are time-consuming and hardware-unfriendly. In contrast, we novelly propose using a static approach to further enhance the role of rotation matrices.

Specifically, we have obtained the activation data $X \in \mathbb{R}^{B \times N \times D}$ of the target layer from CBC, where $B$ is the batch size, $N$ is the number of tokens, and $D$ is the number of channels. First, we observe that the activations form stable statistical characteristics along the batch dimension. We then average $X$ along the batch dimension, resulting in the statistical mean matrix $S \in \mathbb{R}^{N \times D}$. Moreover, during the process of image or video generation, the total number of tokens in DiT is fixed.

Following the approach of DuQuant (Lin et al., 2024), we utilize an outlier metric to assess the outliers across different channels. Specifically, we define the outlier metric for each channel as the absolute maximum value of the activations in that channel. We then sort the outlier metrics in descending order and construct a column-swapping matrix: the top 50% of channels are moved to the left side, and the bottom 50% to the right side. Thus, the left side consists of high-peak channels, while the right side consists of low-peak channels. We also experimented with other metrics, such as variance and range, but the final experimental results showed no significant differences (See ablation part). Therefore, we adopted the simplest form, *i.e.,* the absolute maximum value.

Next, we construct an orthogonal rotation matrix, which is a block Hadamard matrix $\mathbf{H}$, and multiply it by the column-swapping matrix $\mathbf{S}$ to obtain the total orthogonal transformation matrix $\mathbf{G} = \mathbf{SH}$. The Hadamard matrix could effectively smooth the activation matrix by rotating the outliers into smooth parts. Moreover, the column-swapping matrix $\mathbf{S}$ further improves the smoothing ability.

$\mathbf{G}$ is stored as part of the quantization parameters and is invoked during inference. As $\mathbf{G}$ is an orthogonal matrix, *i.e.,* $\mathbf{GG}^T = \mathbf{I}$, it brings lossless smoothing. For weights, $\mathbf{G}$ can be directly absorbed into the weights before quantization by $\hat{\mathbf{W}} = Q(\widetilde{\mathbf{W}}) = Q(\mathbf{G}^T\mathbf{W})$, which does not incur additional storage overhead. For activations, since the matrix is a block Hadamard matrix, the matrix multiplication can be performed using fast Hadamard multiplication with a time complexity of $\mathcal{O}(n^2 \log n)$, where $n$ is typically in the thousands. Hence, compared to the subsequent matrix multiplication with a time complexity of $\mathcal{O}(n^3)$, this step can be ignored.

As for storage, Hadamard matrices enjoy excellent properties. (1) It can be generated online, saving disk storage. (2) The Hadamard matrices' elements are binary, *i.e.,* $\pm 1$, saving GPU memory. We only need to store the column-swapping matrix $\mathbf{S}$, which can be further compressed into a vector $\mathbf{s}$, with its $k$-th element indexing the column after swapping. Considering $D$ is huge for most of the layers, the storage overhead of $\mathbf{s}$ is usually less than $0.1\%$, which can be safely ignored.

Additionally, in the outlier channel swapping, we do not need an overall descending order because this approach facilitates the multiplication with the block Hadamard matrix. The rotation matrix takes the form of a block diagonal matrix, where each block processes 50% of the columns of the activation. Therefore, we only need to swap the columns with higher outlier values to the same side, without forcing the arrangement into descending order. Moreover, the blocked form also fastens the calculation. More detailed analysis can be found in the supplementary material.

## 3.4 Cross-Layer Parameter Searching

In most existing studies, the quantizer parameters, namely $l$ and $r$, have received little attention, despite their significant impact on quantization outcomes. Recent approaches (Zhao et al., 2025) typically adopt simple strategies such as min–max or percentile bounds. However, such coarse methods often result in severe degradation or even collapse under low-bit quantization. To address this issue, we propose **CLPS**, a cross-layer parameter searching method.

The essence of optimizing quantizer parameters lies in minimizing the adverse effects of rounding and clipping errors on model performance. Directly relying on the final model output for this optimization

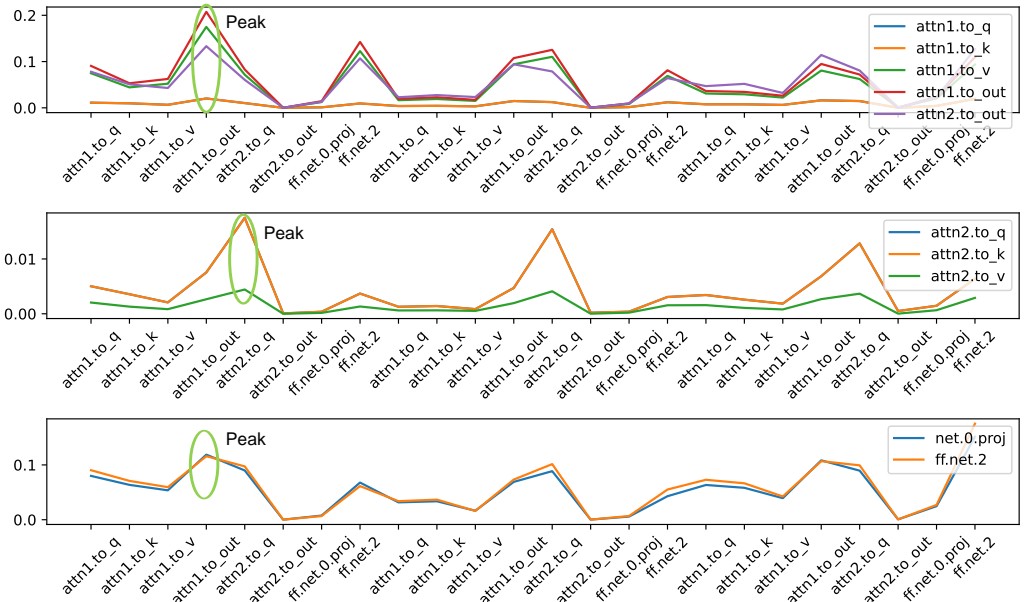

Figure 3: The visualization of cross-layer error when giving a disturbance to stimulate quantization error. For each layer to be optimized, we analyze the most influenced layer, *i.e.,* the peak, with the shift from the original output, and set it as the target layer for quantization parameter optimization.

would be computationally prohibitive, as each layer would require a complete forward pass, and the VAE part needs to be included, which is computationally expensive. On the other hand, using only the local input and output of a given layer would be insufficient to capture its potential influence.

To balance these considerations, our principle is to select, around the layer to be optimized, the subsequent layer exhibiting the largest variance as the target layer. We limit the target layer to be within the subsequent three blocks. To find the target layer, we apply perturbations to the layer to be optimized and perform forward propagation to obtain the output of all the subsequent layers. Perturbations lead to a shift from the original output. We quantify the shift as the L1 norm between the shifted output $\widetilde{y}_{L_t}$ and the original output $y_{L_t}$ of the subsequent layer $L_t$. We determine the target layer as the most influenced layer, which can be written as

$$L_T = \arg\max_{L_t} ||\widetilde{y}_{L_t} - y_{L_t}||_1. \tag{2}$$

Once the optimized layer $L_O$ and the target layer $L_T$ are determined, we perform a grid search over the $l \in \mathcal{S}_l = [Q_\beta(W_{L_O}), Q_\gamma(W_{L_O})]$ and $r \in \mathcal{S}_r[Q_{1-\gamma}(W_{L_O}), Q_{1-\beta}(W_{L_O})]$, where $Q_\eta(W)$ is the $\eta\%$ greatest value of $W$, $\beta$ and $\gamma$ are the grid bounds. The search objective is defined as $l^\star, r^\star = \arg\min_{l \in \mathcal{S}_l, r \in \mathcal{S}_r} ||Y - \hat{Y}_{l,r}||_F^2$, where $Y$ is the original output of $L_T$ and $\hat{Y}_{l,r}$ is the output of $L_T$ after quantizing $L_O$ with $l$ and $r$. With the searched $l^\star, r^\star$, the cross-layer quantization error on the most influenced layer can be minimized to the best extent.

### 3.5 OVERALL

We propose three designs, which are CBC for accurate calibration, OBS for outlier smoothing, and CLPS for determining quantization parameters. Here, we introduce the sequence of these three designs. Overall, we perform CBC, OBS, and CLPS iteratively across Transformer blocks. When quantizing the $k$-th Transformer block $B_k$, the previous blocks are already properly quantized. We first perform CBC to collect the calibration data. Then, we quantize the layer inside the Transformer block one by one. Given a layer to be optimized $L_O$, we first determine the rotation matrix $\mathbf{G}$ in OBS and merge it into the weight matrix. Then, we find the corresponding target layer $L_T$ and perform CLPS to obtain $l^\star$ and $r^\star$ for $L_O$. For the last Transformer block, the target layer is set to be the output of the block. The pseudocode is provided in the supplementary materials.

## 4 EXPERIMENTS

### 4.1 EXPERIMENTAL SETTINGS

**Video Generation Evaluation Settings.** We apply **CLQ** to Open-Sora 1.2 to test the video generation task. All the videos are generated with 100 steps with a CFG scale of 4.0. The evaluation of **CLQ** is

Table 1: Ablation study on our proposed designs, including CBC, OBS, and CLPS. The results on all metrics demonstrate the effectiveness of the proposed designs.

| Method | Single Object | Two Object | Counting | Colors | Position | Color Attribute | Total |
|---|---|---|---|---|---|---|---|
| Naive | 0.934 | 0.525 | 0.425 | 0.722 | 0.050 | 0.050 | 0.451 |
| +OBS | 0.978 | 0.525 | 0.425 | 0.917 | 0.200 | 0.250 | 0.549 |
| +CLPS | 0.978 | 0.575 | 0.425 | 0.917 | 0.275 | 0.250 | 0.570 |
| +CBC | 0.978 | 0.625 | 0.425 | 0.944 | 0.300 | 0.275 | 0.591 |

Table 2: Ablation study on our proposed designs, including CBC, OBS, and CLPS. The results on all metrics demonstrate the effectiveness of the proposed designs.

| Method | Single Object | Two Object | Counting | Colors | Position | Color Attribute | Total |
|---|---|---|---|---|---|---|---|
| Abs Max | 0.972 | 0.650 | 0.425 | 0.917 | 0.225 | 0.375 | 0.594 |
| Percentile | 0.978 | 0.600 | 0.450 | 0.944 | 0.275 | 0.250 | 0.583 |
| Top k Mean | 0.972 | 0.650 | 0.425 | 0.917 | 0.225 | 0.375 | 0.594 |
| Range | 0.978 | 0.625 | 0.425 | 0.917 | 0.250 | 0.325 | 0.587 |
| PAR | 0.975 | 0.575 | 0.425 | 0.917 | 0.275 | 0.325 | 0.582 |

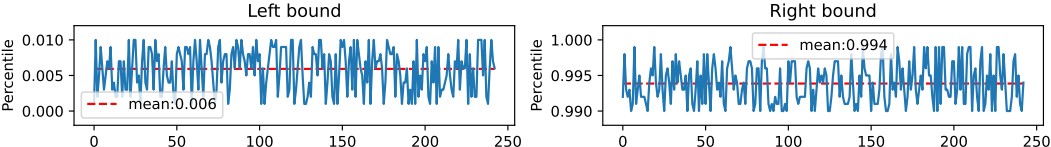

Figure 4: The left and right bound, *i.e.,* quantization parameters of CLPS on PixArt-$\alpha$.

performed on VBench Huang et al. (2024) to provide comprehensive results. Following previous research Zhao et al. (2025), we select 7 major dimensions from VBench, including imaging quality, aesthetic quality, motion smoothness, dynamic degree, background consistency, scene consistency, and overall consistency. The result of the other dimensions can be found in the supplementary material.

**Image Generation Evaluation Settings.** We apply **CLQ** to PixArt-$\alpha$ Chen et al. (2023a) to test the image generation task. When evaluating, the images are generated with a 50-step DPM-solver with the CFG scale of 4.5. We adopt GenEval Ghosh et al. (2023) to evaluate the performance, and all its metrics are reported, including single object, two object, counting, colors, position, attribute binding, and overall. With all these metrics, the performance of our **CLQ** can be thoroughly evaluated.

**Quantization Scheme.** The quantization granularity is set to be per-channel. We test the performance of **CLQ** under three bit settings, including W8A8, W4A8, and W4A4. As the matrices are smoothed enough after OBS, we set the percentile the same for all channels within a matrix in CLPS. We set $\beta = 0, \gamma = 0.01$ for CLPS. To demonstrate the performance of **CLQ**, we select the SOTA PTQ methods for comparison, including ViDiT-Q (Zhao et al., 2025), SmoothQuant (Xiao et al., 2023), QuaRot (Ashkboos et al., 2024), and Q-DiT (Chen et al., 2025).

## 4.2 ABLATION STUDY

We present ablation studies in Tab. 1 to evaluate the contribution of the different components in the proposed **CLQ**. The ablation study is conducted on a subset of GenEval. By randomly choosing the prompts, the generated images are enough to cover different dimensions while requiring less computation. We provide the prompts used in the supplementary material.

**Evaluation of OBS.** As shown in Tab. 1, the proposed OBS could effectively smooth the activation matrix. The quantization error of both weights and activations is reduced 90% compared to those before OBS. Moreover, we also provide a visualization of the activations and weights before and after OBS in the supplementary material, which also supports the obvious smoothing ability of OBS. Therefore, OBS improves the performance of the quantized model with negligible overhead.

**Evaluation of CLPS.** Next, we evaluate the effectiveness of CLPS. After searching for the optimized quantization parameters, the model's performance enjoys obvious improvement. Fig. 3 shows the distribution of the searched parameters along the model depth. The average bound across the whole model is $(0.006, 0.994)$. Both compact results validate the necessity of optimizing the bound.

**Evaluation of CBC.** Next, we incorporate CBC alongside the other designs. The block-wise calibration data provide an accurate guide to OBS and CLPS, further enhancing model performance.

Table 3: Comparison with SOTA PTQ methods on GenEval. The best and the second best results are marked with **bold** and underline, respectively. "-" means the model collapses.

| Method | Bits (W/A) | Single Object | Two Object | Counting | Colors | Position | Color Attribute | Total |
|---|---|---|---|---|---|---|---|---|
| FP | 16/16 | 0.980 | 0.660 | 0.510 | 0.780 | 0.140 | 0.220 | 0.550 |
| Q-DiT | 8/8 | **0.981** | 0.612 | **0.525** | 0.750 | 0.113 | 0.150 | 0.522 |
| SmoothQuant | 8/8 | 0.969 | 0.638 | 0.450 | 0.813 | 0.135 | 0.163 | 0.528 |
| Quarot | 8/8 | **0.981** | **0.663** | 0.438 | 0.788 | 0.135 | 0.175 | 0.530 |
| ViDiT-Q | 8/8 | 0.978 | 0.634 | 0.463 | 0.793 | 0.135 | **0.210** | 0.535 |
| CLQ (Ours) | 8/8 | 0.975 | **0.663** | 0.463 | **0.815** | **0.150** | **0.210** | **0.546** |
| Q-DiT | 4/8 | - | - | - | - | - | - | - |
| SmoothQuant | 4/8 | 0.666 | 0.350 | 0.238 | 0.513 | 0.063 | 0.163 | 0.332 |
| Quarot | 4/8 | 0.772 | 0.613 | 0.363 | 0.588 | 0.088 | 0.175 | 0.433 |
| ViDiT-Q | 4/8 | 0.891 | 0.475 | 0.406 | 0.649 | 0.108 | 0.208 | 0.456 |
| CLQ (Ours) | 4/8 | **0.975** | **0.649** | **0.472** | **0.763** | **0.125** | **0.235** | **0.537** |
| Q-DiT | 4/4 | - | - | - | - | - | - | - |
| SmoothQuant | 4/4 | - | - | - | - | - | - | - |
| Quarot | 4/4 | 0.219 | 0.063 | 0.038 | 0.138 | 0.038 | 0.013 | 0.084 |
| ViDiT-Q | 4/4 | 0.247 | 0.035 | 0.038 | 0.165 | 0.008 | 0.018 | 0.085 |
| CLQ (Ours) | 4/4 | **0.938** | **0.614** | **0.456** | **0.736** | **0.108** | **0.200** | **0.509** |

Table 4: Comparison with SOTA PTQ methods on VBench. The best and the second best results are marked with **bold** and underline, respectively. "-" means the model collapses.

| Methods | Bits (W/A) | Imaging Quality | Aesthetic Quality | Motion Smooth. | Dynamic Degree | BG. Consist. | Scene Consist. | Overall Consist. |
|---|---|---|---|---|---|---|---|---|
| FP | 16/16 | 56.45 | 55.48 | 98.49 | 51.39 | 97.36 | 45.20 | 26.91 |
| Q-DiT | 8/8 | 54.28 | 55.80 | 93.64 | 40.27 | 94.70 | 33.35 | 26.09 |
| SmoothQuant | 8/8 | 55.38 | 55.38 | 98.12 | 37.50 | 97.01 | 36.01 | 26.49 |
| Quarot | 8/8 | 55.37 | 55.38 | 98.20 | 37.50 | 97.26 | 32.31 | 26.59 |
| ViDiT-Q | 8/8 | **55.92** | 54.66 | **98.43** | **50.00** | 97.10 | **41.56** | **26.66** |
| CLQ (Ours) | 8/8 | 55.37 | **55.89** | 98.28 | 41.67 | **97.61** | 36.41 | 26.62 |
| Q-DiT | 4/8 | 23.30 | 29.61 | 97.89 | 4.16 | 97.02 | 0.00 | 4.98 |
| SmoothQuant | 4/8 | 51.47 | 54.87 | 98.11 | 34.72 | 96.76 | 31.35 | 26.22 |
| Quarot | 4/8 | 50.97 | 54.96 | 98.18 | 33.33 | 96.73 | 31.92 | 26.70 |
| ViDiT-Q | 4/8 | 53.16 | 53.04 | **98.28** | **44.44** | 97.30 | 31.82 | 26.29 |
| CLQ (Ours) | 4/8 | **54.79** | **55.81** | 98.18 | 41.67 | **97.75** | **32.85** | **26.71** |
| Q-DiT | 4/4 | - | - | - | - | - | - | - |
| SmoothQuant | 4/4 | 40.41 | 37.82 | 84.63 | **100.00** | 93.81 | 0.00 | 3.57 |
| Quarot | 4/4 | 38.22 | 36.45 | 92.12 | **100.00** | 94.16 | 0.02 | 5.01 |
| ViDiT-Q | 4/4 | 38.52 | 37.72 | 92.71 | **100.00** | 94.70 | 0.44 | 4.67 |
| CLQ (Ours) | 4/4 | **50.69** | **54.14** | **97.42** | 36.11 | **95.94** | **31.69** | **26.33** |

**Choice of Outlier Metric.** In OBS, we need to determine the outlier score of each channel. We test five commonly used metrics, including absolute max value, percentile, top k mean, range, and peak average ratio (PAR). We set the percentile as $(1\%, 99\%)$ and $k = 1\%$ in top k mean. As shown in Tab. 2, there is no evident difference between the tested metrics. Therefore, we select the simplest form, *i.e.,* absolute maximum value.

**Target Layer of CLPS** We observe that the target layer CLPS is stable when considering the model structure. Take OpenSora as an example, the target layer is always the attn1.to_out in the next block for layers in attn1, attn2.to_out, and layers in FFN, as shown in Fig. 3. And it is always the attn1.to_q in the next block for the rest of the layers in attn2.

## 4.3 COMPARISON WITH STATE-OF-THE-ART METHODS

Tab. 3 shows the results on image generation, and Tab. 4 shows the results on video generation. Our proposed CLQ consistently performs the best or the second best on all metrics when it comes to the low-bit scenario. The dynamic degree is an exception. When compressed to W4A4, the generated videos of the compared methods are full of random noise, which brings a high dynamic degree of 100. Therefore, there is a sweet spot for video quality when evaluated on VBench. These outstanding results demonstrate the effectiveness and robustness of the proposed CLQ.

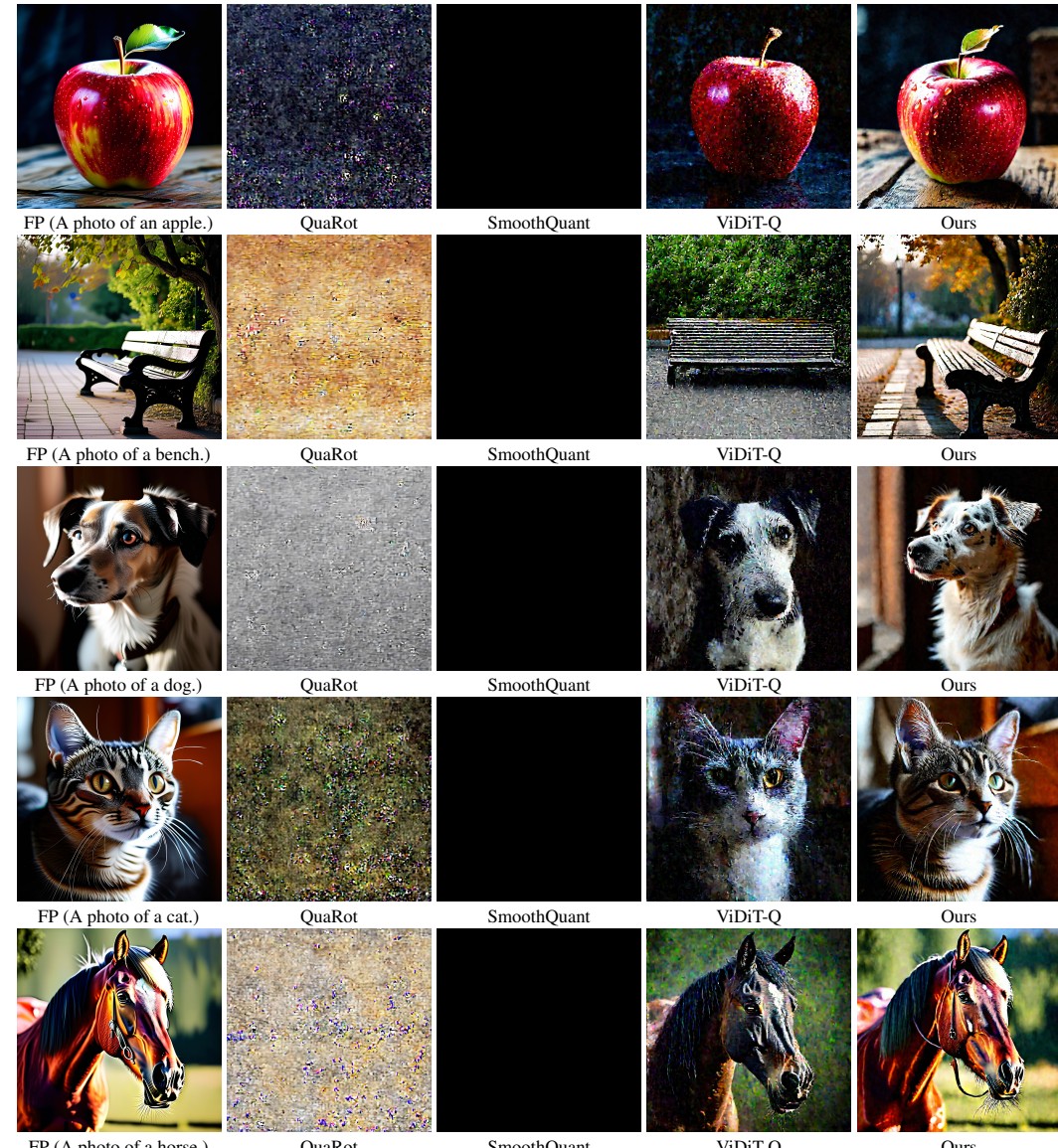

Figure 5: Visual comparison for generation with **W4A4**. We compare our proposed CLQ with current competitive PTQ methods and the full-precision (FP) model. The visual results illustrate that CLQ gains rich details and less noise.

Fig. 5 shows the generated visual content of the proposed CLQ and the SOTA methods under W4A4. SmoothQuant can only generate all black content, while QuaRot generates random noise. ViDiT-Q can keep the semantic content, but obvious noise can be observed around the whole image. In contrast, our results are visually the same as the FP model, representing the excellent performance of the proposed CLQ. More quantitative and qualitative comparison results can be found in the supplementary materials.

## 5  CONCLUSION

We propose CLQ, an efficient post-training method for visual generation models. CLQ consists of three novel designs, including CBC, OBS, and CLPS. CBC provides accurate calibration data for the other two components. OBS leverages the Hadamard matrix to smooth the outliers with negligible overhead. CLPS searches for the quantization parameters with the most influenced subsequent layer. All three designs together enable the visual generation model to provide FP-similar content when compressed to W4A4. Future work will focus on lower bit-width and further improving performance.

## A    ETHICS STATEMENT

The research conducted in the paper conforms, in every respect, with the ICLR Code of Ethics.

## B    REPRODUCIBILITY STATEMENT

We have provided implementation details in Sec. 4. We will also release all the code and models.

## C    LLM USAGE STATEMENT

Large Language Models (LLMs) were used solely for polishing writing. They did not contribute to the research content or scientific findings of this work.

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
