# CLQ: CROSS-LAYER GUIDED ORTHOGONAL-BASED QUANTIZATION FOR DIFFUSION TRANSFORMERS

## A  PIPELINE

The detailed pipeline of the proposed CLQ is shown in Alg. 1

---

**Algorithm 1:** CLQ Process Pipeline

---

**Data:** Dataset $\mathcal{D} = \{X_i\}_{i=1}^N$, DiT model $M$
**Result:** Rotation Matrices $\mathcal{G}$, Quantization Parameters $\mathcal{Q}$.
$\mathcal{G} \leftarrow []$;
$\mathcal{Q} \leftarrow []$;
**foreach** *block $B$ in $M$* **do**
    // CBC
    Perform forward propagation within $B$ with $\mathcal{D}$;
    Record the activation within $B$;
    **foreach** *layer $L_O$ in $B$* **do**
        // OBS
        Calculate the outlier score for layer $L_O$;
        Swap channels for $L_O$ such that the top 50% outlier channels are on the top left 50% ;
        Generate the channel swapping matrix **S** with above step;
        Generate block Hadamard matrix **H**;
        Compute rotation matrix $\mathbf{G} = \mathbf{SH}$;
        $\mathcal{G}$.append(**G**);
        // CLPS
        Apply perturbation to **W** such that it changes to $\hat{\mathbf{W}}$;
        Perform forward propagation with $\mathcal{D}$;
        Record the activations within the three blocks following $B$;
        Calculate the shift from the original activation;
        Find the target layer $L_T$ with the biggest shift;
        Search the optimal quantization bound $l^\star, r^\star = \arg\min_{l \in \mathcal{S}_l, r \in \mathcal{S}_r} ||Y - \hat{Y}_{l,r}||_F^2$;
        $\mathcal{Q}$.append($(l^\star, r^\star)$)
    **end**
**end**
**return** $\mathcal{G}, \mathcal{Q}$;

---

## B  EFFICIENCY OF HADAMARD-BASED MATRIX MULTIPLICATION

The computational advantage of employing Hadamard matrices arises from their close connection to the Fast Walsh–Hadamard Transform (FWHT) Hamood & Boussakta (2011). Unlike general dense matrices, whose multiplication with an $n \times n$ matrix requires approximately $2n^3 - n^2$ floating-point operations (flops), the Hadamard matrix consists only of entries $\pm 1$ and admits a recursive structure that enables efficient evaluation. Specifically, multiplying an $n \times n$ Hadamard matrix with a vector can be carried out in $n \log_2 n$ flops using FWHT, since the operation decomposes into $\log_2 n$ stages of pairwise additions and subtractions, entirely avoiding multiplications. Consequently, when multiplying a Hadamard matrix with an arbitrary $n \times n$ matrix on the appropriate side (e.g., $HA$), the total cost reduces to $n^2 \log_2 n$ flops, compared to cubic complexity for standard matrix–matrix multiplication. Moreover, by exploiting the identity $(AH)^T = HA^T$ and the symmetry of Hadamard

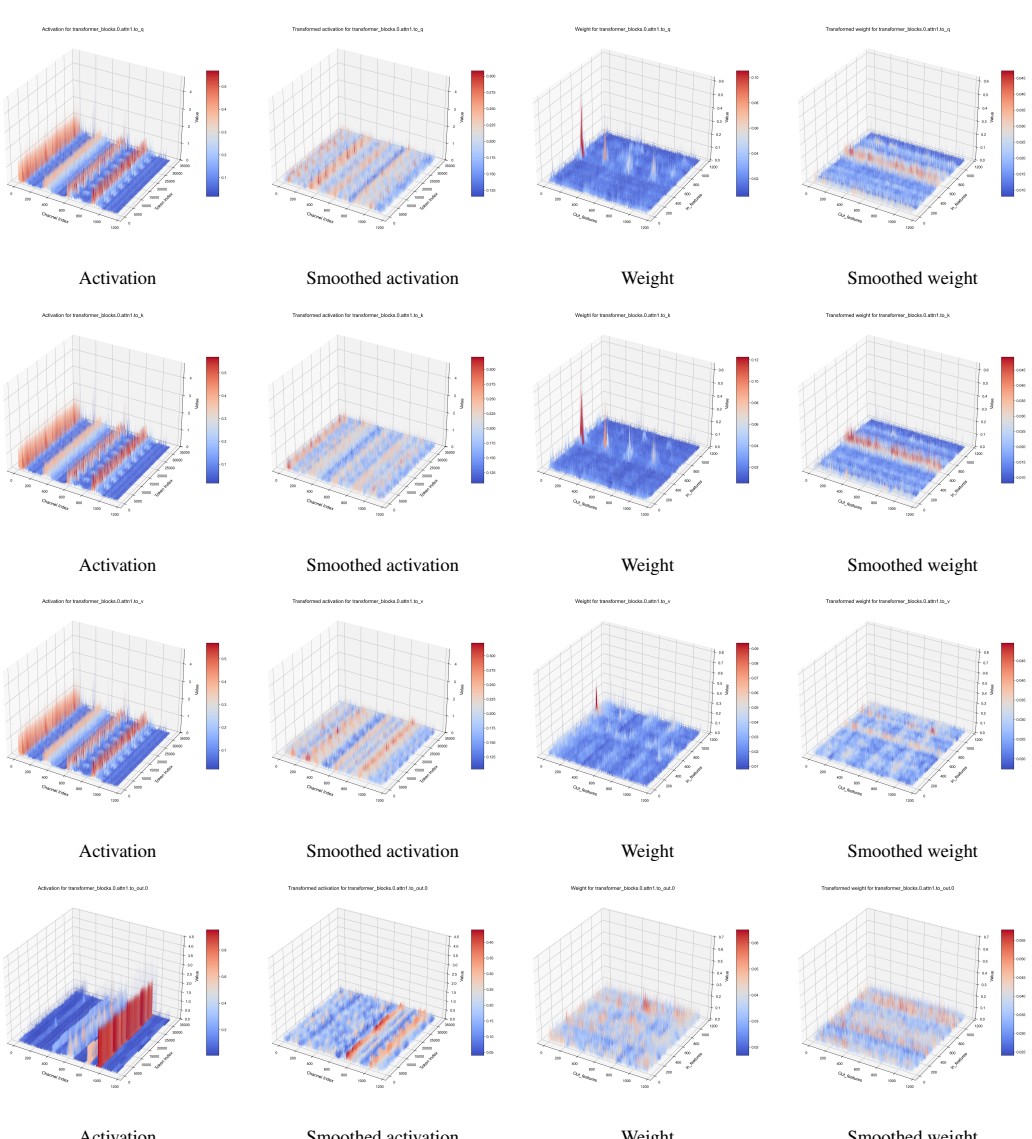

Figure 1: Visualization of the activation and weight before and after CLQ.

matrices, the same acceleration can be leveraged for $AH$ as well, yielding an overall reduction from $O(n^3)$ to $O(n^2 \log n)$.

## C    SMOOTHED WEIGHT AND ACTIVATION

We visualize the weight and activations in Fig. 1. To demonstrate the outliers, the displayed values are the absolute value of the original value. All the activations are smoothed after CLQ, while most of the weights are smoother after CLQ. This visualization demonstrates the outstanding performance of the proposed OBS.

## D    VBENCH

We provide results on other dimensions of VBench in Tab. 1.

Table 1: Comparison with SOTA PTQ methods on VBench. The best and the second best results are marked with **bold** and underline, respectively. "-" means the model collapses.

| Methods | Bits (W/A) | Subject Consist. | Temporal Flickering | Object Class | Multiple Objects | Human Action | Color | Spatial Relationship | Temporal Style | Appearance Style |
|---|---|---|---|---|---|---|---|---|---|---|
| FP | 16/16 | 93.71 | 99.47 | 76.98 | 47.56 | 91.00 | 75.74 | 44.78 | 25.51 | 24.15 |
| Q-DiT | 8/8 | 86.94 | 99.41 | 78.52 | 42.85 | 88.00 | 67.56 | 54.10 | 23.63 | 24.13 |
| SmoothQuant | 8/8 | 93.20 | 99.42 | 81.38 | 44.58 | 89.00 | 69.04 | 54.73 | 25.62 | 24.04 |
| Quarot | 8/8 | 93.32 | 99.41 | 80.17 | 41.92 | 91.00 | 67.90 | 52.10 | 25.56 | 24.07 |
| ViDiT-Q | 8/8 | 93.59 | 99.20 | 76.66 | 47.56 | 89.00 | 71.19 | 48.48 | 25.57 | 24.05 |
| CLQ (Ours) | 8/8 | 93.62 | 99.49 | 81.52 | 52.88 | 91.00 | 71.89 | 54.96 | 25.63 | 24.15 |
| Q-DiT | 4/8 | 91.51 | 97.42 | 65.14 | 38.47 | 79.00 | 62.31 | 48.28 | 22.31 | 17.38 |
| SmoothQuant | 4/8 | 92.74 | 99.40 | 74.28 | 53.12 | 86.00 | 68.25 | 50.71 | 24.62 | 24.13 |
| Quarot | 4/8 | 92.91 | 99.38 | 74.92 | 46.04 | 86.00 | 73.37 | 50.58 | 25.33 | 23.50 |
| ViDiT-Q | 4/8 | 92.87 | 99.28 | 78.00 | 45.42 | 88.00 | 74.23 | 52.08 | 25.60 | 24.09 |
| CLQ (Ours) | 4/8 | 93.61 | 99.56 | 82.04 | 56.33 | 91.00 | 72.35 | 53.61 | 25.24 | 24.12 |
| Q-DiT | 4/4 | - | - | - | - | - | - | - | - | - |
| SmoothQuant | 4/4 | 79.82 | 86.71 | 2.13 | 0.00 | 1.00 | 0.00 | 0.00 | 2.94 | 20.67 |
| Quarot | 4/4 | 87.27 | 94.58 | 5.30 | 0.00 | 1.00 | 0.00 | 0.03 | 4.36 | 20.91 |
| ViDiT-Q | 4/4 | 88.19 | 94.72 | 5.93 | 0.00 | 0.00 | 0.00 | 0.11 | 4.08 | 20.84 |
| CLQ (Ours) | 4/4 | 90.29 | 98.56 | 72.47 | 41.46 | 79.00 | 65.97 | 49.14 | 25.85 | 24.19 |

# E  VISUAL COMPARISON

In this section, we provide more visual comparison in Fig. 2, 3, 4, and 5 with SOTA method under various bit settings.

## REFERENCES

Monir T Hamood and Said Boussakta. Fast walsh–hadamard–fourier transform algorithm. *IEEE TSP*, 59(11):5627–5631, 2011.

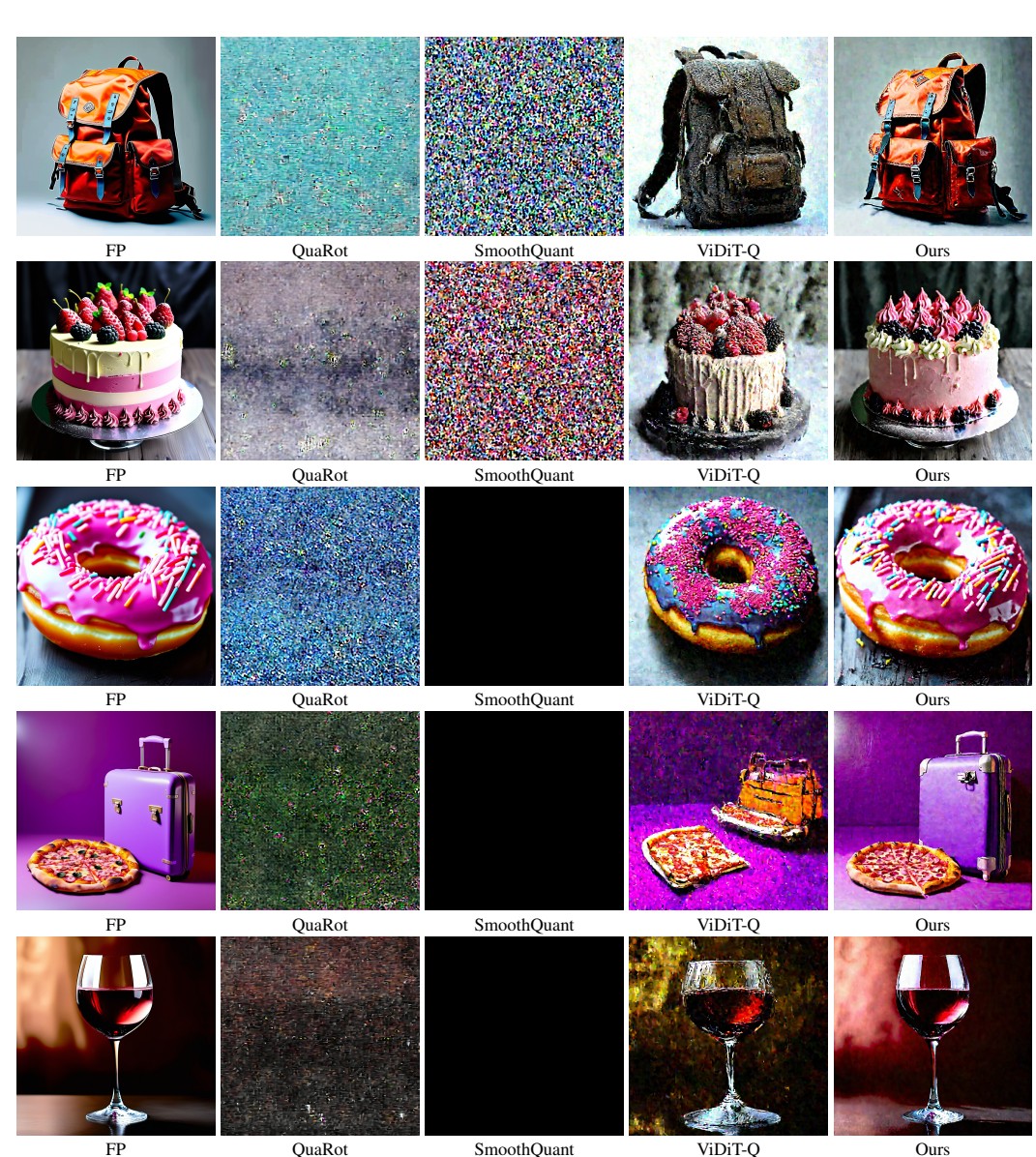

Figure 2: Visual comparison for image generation with **W4A4**.

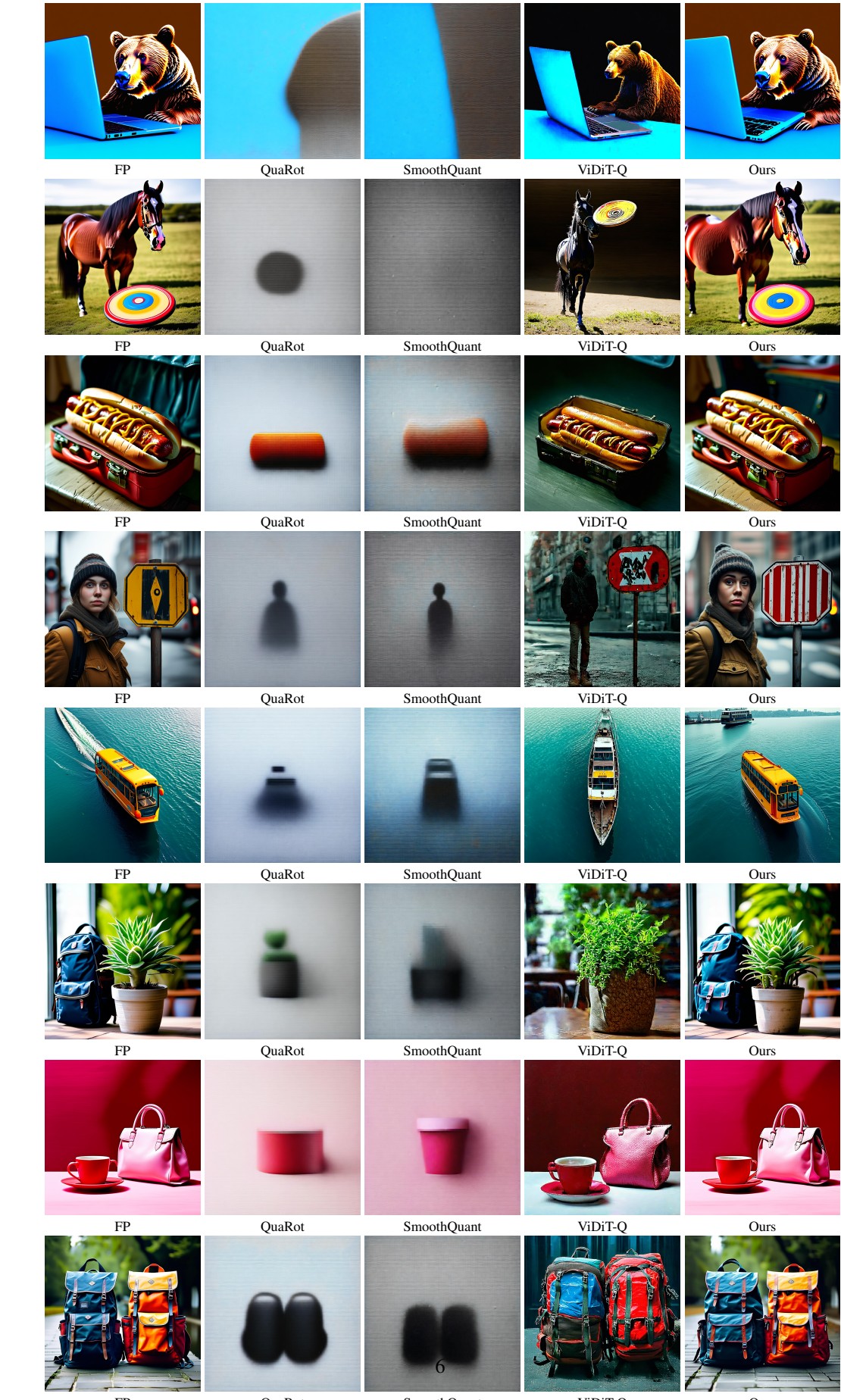

Figure 3: Visual comparison for image generation with **W4A8**.

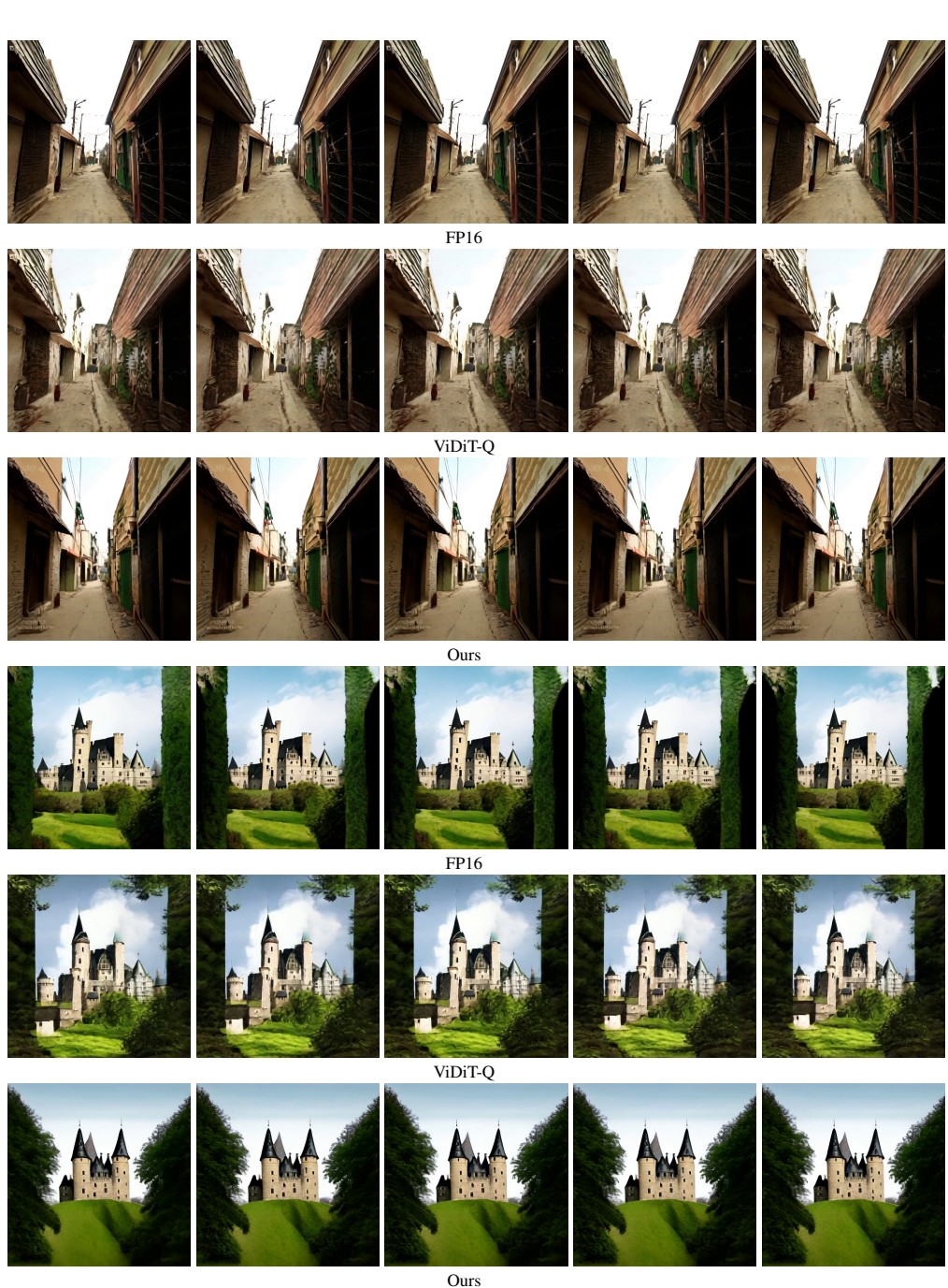

Figure 4: Visual comparison for video generation with **W4A8**.

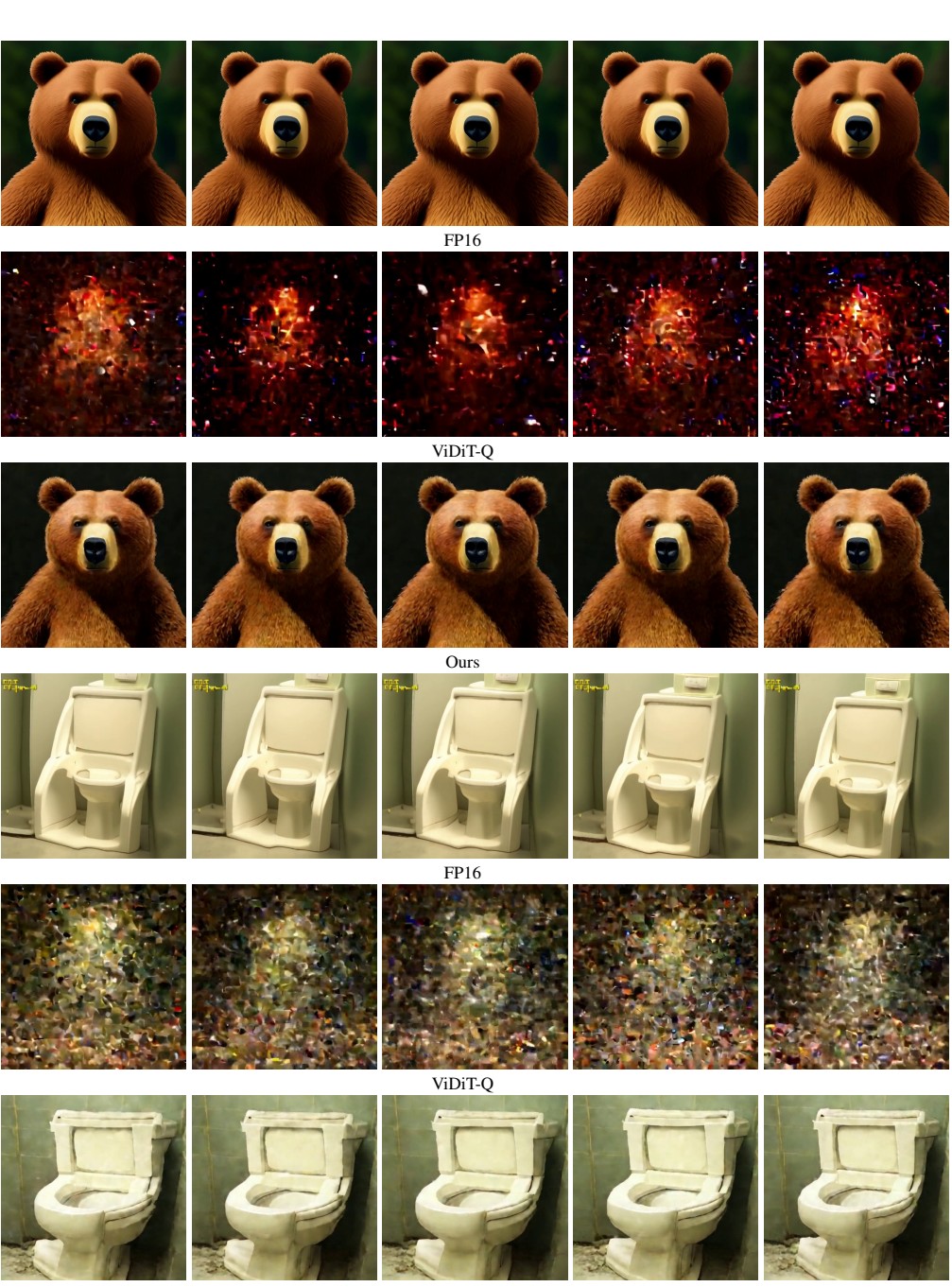

Figure 5: Visual comparison for video generation with **w4a4**.