# OpenReview forum: "CLQ: Cross-Layer Guided Orthogonal-based Quantization for Diffusion Transformers"
_ICLR.cc/2026/Conference — Submitted to ICLR 2026_

### Official Review · Reviewer_szxA · 2025-10-15

**Soundness:** 2
**Presentation:** 2
**Contribution:** 2
**Rating:** 2
**Confidence:** 4

**Summary:**

Authors propose CLQ, a method that addresses outliers in DiT-based diffusion transformers, in turn improving quantization. They claim three novel contributions as part of CLQ:
1. A novel cross-block calibration method that can reduce quantization error across blocks
2. An "orthogonal-based smoothing" method, that groups channels into blocks of similar magnitudes, before applying Hadamard to each block
3. "Cross-layer parameter searching" for setting quantization parameters

They show W4A4 compression with low deterioration and claim SOTA.

**Strengths:**

The paper has clear strengths:
* Important problem, from both a cost, energy, and latency perspective, for which even small improvements can have a massive impact
* W4A4 performance is promising
* method possibly lighter than methods that use end-to-end training

**Weaknesses:**

**1. Contribution.**

I have some doubts about the contributions.

**CBC.** It makes sense to take earlier errors into account. However, it seems to me odd that the input *and* the output is changed. Would it not make more sense to keep the target of each block (the FP output of the full FP model) the same, but only change the input (i.e. to the quantized models' previous layers' output)? Now, it seems to me that actually there could still be error quantization---e.g. if previous blocks have some error, it would make sense that the new block would try to "correct" for this error (instead of trying to match the FP block's output given the error as input).

**OBS**. I'm not sure I understand OBS and am not convinced by the novelty claim of OBS. Could the authors elaborate on the advantage of the permutation, considering the Hadamard mixes across all channels anyway? Also, the authors write:
>  [L224] However, previous studies (Lin et al., 2024; Ashkboos et al., 2024) typically adopted dynamic
approaches to construct the rotation matrix, which are time-consuming and hardware-unfriendly. In
contrast, we novelly propose using a static approach to further enhance the role of rotation matrices.

I'm unsure what the authors mean, exactly. QuaRot (Ashkboos et al) use Hadamard transforms, which are completely static. Other works (e.g. SpinQuant, FlatQuant, OstQuant, FPTQuant) train some transformations, but they are also completely fixed during inference. The "OBS" method seems to just consist of a permutation (e.g. used in DuQuant) and Hadamard transform (used by almost everyone), so could the authors please elaborate on what the contribution is?

**CLPS.** This seems interesting, but I'm not sure this is better than end-to-end training or even per-block learning of activation and weight clipping (e.g. FlatQuant). Authors state
> [L269] Directly relying on the final model output for this optimization would be computationally prohibitive, as each layer would require a complete forward pass, and the VAE part needs to be included, which is computationally expensive.

This argument is incorrect---all layers can be trained at the same time. End-to-end training has been used successfully in e.g. SpinQuant, OstQuant, FPTQuant. It requires just a full forward and backwards pass through the model to update all layer weights, not as authors claim a full forward pass for each *layer* independently. There may still be an advantage of a block-by-block approach (e.g. to keep only one block in memory and increase the batch size), however, the current approach is very expensive---for each layer $L_O$, we require multiple forward passes through that layer *and the next layers*, **and** we need to do this for the different quantization grids of layer $L_O$. This seems quite an elaborate method, that would scale poorly for large candidate grids (i.e. large $S_r$ and $S_l$). It is also unclear how this method would work when there are many quantizers in each block (including channel-wise scaling using different quantizers for each row)---do we find a grid for each quantizer in the block independently? This would scale exponentially in the number of quantizers per block.

**2. Experimental details**

There are not enough details to understand the results section. For Section 4.2 (Table 1), what quantization setting is used (W4A4?), and how is the Naive baseline implemented (e.g. no transforms at all?)? What data is used for calibrating CLPS and CBC? What is the cost of calibration compared to other methods?

**3. Insufficient evidence for contributions**

The results look good, but I'm concerned that evidence for the contributions is insufficient.
1. There are only video tasks, not image, which make it difficult to compare to literature numbers. The Table 3 results for ViDiT-Q don't match their paper's.
2. Authors also ignore and do not discuss transforms/rotations with learneable parameters, e.g. SpinQuant, FlatQuant, SVDQuant, that perform vastly better than e.g. QuaRot.
3. I very much appreciate the current paper's ablations, but it would have been very useful to also have a direct comparison with other methods. E.g., (1) OBS vs standard Hadamard vs some learnable transform (e.g DuQuant, but preferably FlatQuant), (2) end-to-end training vs CLPS+CBC vs just per block training, (3) cost of training CLPS+CBC vs cost of training end-to-end vs per-block training but learnable transforms (preferably FlatQuant). I understand you can't do everything under the sun, but I currently do not see enough evidence that OBS is better than existing (trainable) transforms, and that CLPS+CBC is better/cheaper than some light QAT (in particular, per-block training of transforms or short end-to-end training)


I'm very much willing to increase my score if the authors address my concerns.

**Questions:**

See Weaknesses. In particular,
* How does CLQ compare to methods that train transforms (e.g. FlatQuant or OstQuant)
* How expensive is CLQ to calibrate?

---

### Official Review · Reviewer_nt1d · 2025-10-24

**Soundness:** 2
**Presentation:** 2
**Contribution:** 2
**Rating:** 2
**Confidence:** 4

**Summary:**

This paper introduces CLQ as a combination of three post-training quantization steps aiming to improve performance on low bit width regimes. Namely the paper introduces Cross Block Calibration (CBC) to collect more accurate calibration data, Orthogonal-Based Smoothing (OBS) to better spread the outlier channels and Cross-Layer Parameter Search (CLPS) as a local procedure to tune the quantization hyper-parameters.
Overall the combination of these steps improves the image and video generation quality on the Open-Sora and PixArt-$\alpha$ architectures, with significant improvements especially in the W4A4 regime.

**Strengths:**

* The paper introduces a series of changes for the PTQ procedure of DiT architectures that result in substantial improvements in low bit width settings.

* Figure 2 does a good job in summarizing the proposed method, making the general idea clear from the beginning.

**Weaknesses:**

* Some sections of the paper are not clearly described, namely:
   * The procedure to generate $S$ and $H$ in lines 242-261 is quite ambiguous.
   * The plots in Figure 3 are not entirely described.  Without an explanation of the quantities in the legend and on the x and y axis, it is quite challenging to interpret.
* The paper proposes 3 variations of standard PTQ procedures that include many hyper-parameters. However, the paper does not provide much intuition or motivation regarding each choice. The ablations included in the paper are also not covering the impact of each suggested modification in detail.
* Some crucial details required to interpret and reproduce the results are missing from the main text (see questions).

**Questions:**

1. What is the intuition behind swapping the channels so that the ones with outliers are mixed together? Is this procedure more effective than doing the opposite (i.e. making sure that the outliers are distributed uniformly in each block to spread them across more channels)? How does OBS compare to vanilla block Hadamard (no S) or Hadamard with random permutations (randomized Hadamard)? How does this change for different block size?


2. The paper mentions that, considering the limited depth of one transformer block, during calibration, only the previous block is quantized. What is the reason behind this choice? Does it reduce the calibration time? How much of an impact does it make compared to quantizing all previous blocks instead in terms of the end2end model performance?



3. Can the authors clarify the reasons behind the choice of using the layer with the largest variance located at most 3 blocks from the current one when setting the quantization range? Why is the L1 norm of the quantization a good target for this procedure? Does this procedure favor layers with larger norm (since the error is also scaled up)?  Can the authors provide more intuition behind this choice and a comparison with block-wise optimization?

4. How much time/compute does the proposed CLQ calibration procedure require compared to the baselines included in this work? Which layers and activations are quantized in each model reported in Table 3 and 4? How is the 3.95x speedup reported in the abstract estimated?

---

### Official Review · Reviewer_tS79 · 2025-10-31

**Soundness:** 2
**Presentation:** 2
**Contribution:** 2
**Rating:** 2
**Confidence:** 4

**Summary:**

This paper proposes a cross-layer guided orthogonal-based quantization method for DiTs. The approach focuses on optimizing calibration data, rotation matrices, and cross-layer parameter search to improve W4A4 quantization performance.

**Strengths:**

- The proposed OBS module is shown to be effective through ablation studies.
- The visualization results are informative and clearly presented.

**Weaknesses:**

- The related work section omits several important DiT quantization baselines, such as SVDQuant, PTQ4DiT, and QDiT. These baselines are also missing from the experimental comparisons.
- The paper claims a 3.98× memory saving and 3.95× speedup in the abstract and introduction, but no inference results are provided in the main text or appendix to support these claims.

**Questions:**

- Since CBC operates online, what is the additional inference cost introduced in terms of latency?
- As the proposed method is inspired by DuQuant, how does its performance compare directly with DuQuant under similar settings?
- Could you clarify the setup used for memory usage and speedup measurements? Also, how do these results compare with baselines such as ViDiT-Q?

---

### Official Review · Reviewer_CWXF · 2025-11-01

**Soundness:** 3
**Presentation:** 3
**Contribution:** 2
**Rating:** 4
**Confidence:** 4

**Summary:**

This paper proposes a cross-layer guided orthogonal-based quantization method for DiTs. Three key techniques are introduced: cross-block calibration, orthogonal-based smoothing, and cross-layer parameter searching. Experiments were conducted on both image and video generation tasks and demonstrate good results on W4A4 settings.

**Strengths:**

1. The performance at 4-bit (W4A4) is excellent.
2. Experiments were conducted on both image generation and video generation models, demonstrating the method's generalizability.

**Weaknesses:**

1. The first contribution (cross-block calibration) is a standard practice in LLMs (e.g., GPTQ). It is less common in diffusion models because diffusion calibration requires multi-timestep data (i.e., running a full forward pass, e.g., 50 steps, for each sample). In this paper, quantizing each transformer block involves re-executing this sampling step, which I suspect is extremely slow. The paper also lacks an analysis of algorithmic efficiency.

2. Similarly, the third contribution (cross-layer parameter searching) likely involves the multi-timestep issue. To determine the output error of subsequent layers, is it also necessary to perform a multi-step forward pass?

3. The specific quantizer settings (e.g., granularity, symmetric/asymmetric) are not specified in the main text.

4. The models tested seem somewhat outdated and few in number. I suggest the authors refer to other related works and conduct tests on models such as wan, cognex, and flux.

5. More comparison methods need to be included, such as DVDQuant[1] and SVDQuant[2], which are both post-training methods.

6. The quantization settings for the ablation study are not specified. Furthermore, the "naive method" baseline is not defined, which is very confusing. Given the excellent W4A4 results, why not provide the ablation study under W4A4 settings?

7. Regarding the special orthogonal matrix used in the paper, its latency overhead compared to other orthogonality-based methods is not discussed.

[1] DVD-Quant: Data-free Video Diffusion Transformers Quantization

[2] Svdquant: Absorbing outliers by low-rank components for 4-bit diffusion models

**Questions:**

Please see the weaknesses above.

---

### Meta-Review · Area_Chair_U9Tw · 2025-12-22

**Summary:**

The paper proposes a post-training quantization method for Diffusion Transformers (DiTs). The authors aim to achieve high-quality results at low bit-widths. While the reviewers acknowledged the performance of the proposed method and the generalizability across image and video tasks, they still lean towards rejection with the concerns of missing baselines and comparisons, limited technical novelty, and the issues around efficiency and reproducibility.

**Reviewer Concerns:**

The authors did not provide rebuttal.

**Reviewer Scores:**

The authors did not provide detailed rebuttal to each reviewer, so the reviewers were not involved in the discussion.

---

### Decision · Program_Chairs · 2026-01-26

Reject